# Characterization of a Human Gastrointestinal Stromal Tumor Cell Line Established by SV40LT-Mediated Immortalization

**DOI:** 10.3390/ijms241713640

**Published:** 2023-09-04

**Authors:** Xiangchen Hu, Peng Su, Bo Liu, Jingwei Guo, Zitong Wang, Cai He, Zhe Wang, Youwei Kou

**Affiliations:** 1Department of General Surgery, Shengjing Hospital of China Medical University, Shenyang 110004, China; 13940131435@yeah.net (X.H.);; 2Medical Research Center, Shengjing Hospital of China Medical University, Shenyang 117005, China; 3Department of Pathology, Shengjing Hospital of China Medical University, Shenyang 110004, China

**Keywords:** gastrointestinal stromal tumor, immortalization, SV40LT, cell lines, imatinib

## Abstract

Gastrointestinal stromal tumors (GISTs) are the most common mesenchymal tumors in the digestive tract and originate from the interstitial cells of Cajal (ICC), which is the pacemaker for peristaltic movement in the gastrointestinal tract. Existing GIST cell lines are widely used as cell models for in vitro experimental studies because the mutation sites are known. However, the immortalization methods of these cell lines are unknown, and no Chinese patient-derived GIST cell lines have been documented. Here, we transfected simian virus 40 large T antigen (SV40LT) into primary GIST cells to establish an immortalized human GIST cell line (ImGIST) for the first time. The ImGIST cells had neuronal cell-like irregular radioactive growth and retained the fusion growth characteristics of GIST cells. They stably expressed signature proteins, maintained the biological and genomic characteristics of normal primary GIST cells, and responded well to imatinib, suggesting that ImGIST could be a potential in vitro model for research in GIST to explore the molecular pathogenesis, drug resistance mechanisms, and the development of new adjuvant therapeutic options.

## 1. Introduction

Gastrointestinal stromal tumors (GISTs) are the most common mesenchymal tumors in the digestive tract that originate from gastrointestinal pacemaker cells (i.e., interstitial cells of Cajal, ICC) or related stem cells [1,2]. GISTs are typically driven by mutations in the receptor tyrosine kinase oncogene (C-KIT) or the platelet-derived growth factor receptor α (PDGFRα), which account for 85% or 5–10% of all cases of GIST, respectively [3,4,5]. GISTs without C-KIT or PDGFRα mutations are considered wild-type GISTs (WT-GISTs) [6]. Both molecular analysis for C-KIT/PDGFRα mutations and immunohistochemistry for CD117/DOG-1 protein expression are the gold standards for the diagnosis of GISTs [7]. Currently, surgery is the first-line treatment for resectable and metastasis-free GISTs [8,9]. Patients presenting with metastatic disease should receive tyrosine kinase inhibitor (TKI) therapy as the initial treatment rather than undergo upfront surgery [10]. Imatinib mesylate, a selective TKI that targets C-KIT and PDGFRα, is used in the treatment of unresectable or metastatic GISTs and can significantly improve the five-year survival rates of patients [11,12]. Unfortunately, approximately 50% of patients develop secondary resistance after two years of treatment, which is called imatinib secondary resistance [13,14].

Basic and translational research on GIST requires appropriate experimental models, such as cells, animals, and organoids [15]. Cell models of GISTs mainly include primary cells and immortalized cell lines, which are very difficult to culture [16]. The existing GIST cell lines include GIST882, GIST-T1, GIST430, GIST48, GIST62, and GIST552 [17,18,19] and are widely used as models for in vitro experimental studies of GISTs because of the availability of mutation site information [19,20]. However, immortalization methods for these cell lines have not been reported, and no data on GIST cell lines have been published, except for that of GIST882 [20]. Transduction of simian virus 40 large T antigen (SV40LT) is one of the most common methods for primary cell immortalization, and its mechanisms are well documented [21,22]. SV40LT transforms cells into the S phase and prolongs the cell cycle by inhibiting the functions of P53 and Rb. In addition, SV40LT maintains the telomere length through secondary alterations of telomerase activity [23,24,25]. To date, no human GIST cell lines have been reported to be developed by immortalization with SV40LT.

Here, we established an immortalized human GIST cell line (ImGIST) by transfecting SV40LT into Chinese patient-derived GIST cells. ImGISTs were continuously cultured in vitro for six months and more than 35 passages. Follow-up studies characterized the full range of ImGISTs and described their morphological, cytogenetic, and biological characteristics, and comparative analyses with primary GISTs were performed. These results suggest that ImGIST can be a potential in vitro model to explore molecular pathogenesis and drug resistance mechanisms, as well as develop new adjuvant therapeutic options for GISTs.

## 2. Results

### 2.1. Pathological Assessment of the GIST Tumor Tissue

The hematoxylin-eosin (HE) staining of the GIST tumor tissue was consistent with typical GIST pathology. The cells were spindle-shaped and arranged in bundles, while the mitotic count per 50 high-power fields (HPFs) was approximately 1. Immunohistochemical results showed that: DOG-1 (+); Ki-67 (3% +); CD34 (+); CD117 (+); Desmin (−); S-100 (−); SMA (−); SDHB (+); and pan-TRK (−) (Figure 1).

### 2.2. Primary Cell Culture and Identification of Primary GIST Cells

We successfully isolated and purified primary GIST cells (PriGISTs) from the gastric stromal tumor tissue of a patient. The PriGIST cells were irregularly rod-shaped after isolation (Figure 2A), and after 24 h, the cells adhered to the wall, with approximately 90% of the cells initially showing spindle-shaped morphology (Figure 2B). The cells appeared to proliferate slowly and increased in size and deformity at passage 12 (P12) (Figure 2C). The PriGIST cells in P2 were positive for DOG-1, CD34, and CD117, which was consistent with the immunohistochemical results of the GIST pathological tissue. (Figure 2D–F).

### 2.3. ImGIST Cell Line Characterization

#### 2.3.1. ImGIST Cell Morphology

The ImGISTs cells had a shorter spindle-shaped morphology than that of the PriGISTs cells and irregular neuronal cell-like radial growth, while the fused growth characteristic associated with PriGISTs remained (Figure 2I).

#### 2.3.2. Genetic Mutation Analysis of the ImGIST Cell Line

An identical *C-KIT* 11 V560del mutation was detected in ImGIST and PriGIST cells by Sanger sequencing (Figure 3A), and the mutation site was the same as that in the GIST tissue.

#### 2.3.3. Short Tandem Repeat Analysis of ImGISTs

The match ratio between ImGIST and PriGIST cells was 100% according to the short tandem repeat (STR) analysis at 21 different loci (Table 1). These results indicated that ImGIST represents a new immortalized human GIST cell line. The cell typing results were good, and no multiple alleles or human cross-contamination was found.

#### 2.3.4. Karyotype Analysis of ImGIST Cells

Microscopic analysis of 20 mid-karyotypes revealed that ImGIST cells were hypodiploid, with a representative karyotype of 44–45, XX. +3, del(6)(q13), der(6;12)(p10;q10), add(7)(p13), del(12)(p11), add(12)(p11), −14, −22. The ImGIST cell line had clonal deletions of the long arm of chromosome 6, short arm of chromosome 12, and the typical deletions of chromosomes 14 and 22 (Figure 3B).

#### 2.3.5. Effects of Imatinib Inhibition on ImGIST Cell Growth

We detected a consistent *C-KIT* 11 mutation in PriGIST and ImGIST cell lines. Imatinib is the first-line drug for the treatment of GISTs with *C-KIT* 11 mutations [10]. We used imatinib-treated ImGIST cells to evaluate their drug sensitivity, which revealed that the ImGIST cells were sensitive to imatinib, with a half maximal inhibitory concentration (IC50) of 29.63 ± 1.80 μM (Figure 4A). After treatment with imatinib at IC50 for 24 h, most ImGIST cells shrank, had small spindle-shaped morphology, and lost their fusion growth characteristics, while a few cells died and floated into the culture medium (Figure 4B).

#### 2.3.6. Effects of Different Serum Concentrations on ImGIST Cell Growth

The proliferation of ImGIST cells was dependent on the concentration of fetal bovine serum (FBS), with a 48 h culture further increasing the cell number. These results suggested that 15% FBS was the optimal concentration for the maintenance of ImGIST cells in culture (Figure 4C).

### 2.4. Comparison between the Biological Characteristics of ImGIST and PriGIST Cell Lines

#### 2.4.1. Cell Growth Analysis

Cell growth was assessed using the CCK-8 assay. The ImGIST cells proliferated faster and entered the logarithmic growth phase more quickly than PriGIST cells, which exhibited extremely limited growth (*p* < 0.001) (Figure 4D). After stable ImGIST cell growth, the cell population doubling time (Td) was calculated. The mean Td values for ImGIST and PriGIST cells were 37.8 h and 74.1 h, respectively (*p* < 0.01) (Figure 4D).

#### 2.4.2. Cell Cycle

The percentage of ImGIST cells was significantly higher than the percentage of PriGIST cells (P35) in the S phase (29.08 ± 7.91% vs. 43.16 ± 4.98%, *p* < 0.05) (Figure 5A–C), which indicated that ImGIST cells had remarkably active DNA synthesis compared to the PriGIST cells. The proliferation indices (PI) of ImGIST cells (P35) and PriGIST cells were 0.38 ± 0.06 and 0.61 ± 0.02, respectively (*p* < 0.001) (Figure 5D), indicating that the proliferation rate of the PriGIST cells increased significantly after immortalization.

#### 2.4.3. SV40LT, P53, and CD1117 Expression Analysis

Real-time PCR and western blotting revealed that SV40 mRNA or protein was not expressed in PriGIST cells but was highly expressed in all generations of ImGIST cells, with no significant difference between them (Figure 6A,D). The expression levels of mRNA and protein of P53 were significantly higher in PriGIST cells than in ImGIST cells, and there were no significant differences in these levels between ImGIST generations (Figure 6B,E). Western blotting results showed that the expression of CD117, a signature protein of GIST cells, was not significantly different between PriGIST and ImGIST cells in each generation (Figure 6F).

## 3. Discussion

The morphology and characteristics of primary GIST cells isolated from tissues were almost identical to their source cells [26]. However, these cells cannot be expanded during in vitro culture for long periods because of the stable and long-term loss of their proliferative capacity, although they are still viable and metabolically active [27]. Our several experiments have shown that primary GIST cells exhibit drastic morphological changes and growth retardation during in vitro culturing for approximately 10 generations and are unable to undergo spontaneous immortalization. Therefore, primary GISTs may not be an ideal cell source for experimental studies or at least not a stable cell source for long-term experiments.

Given that immortalized cell lines can pass multiple times in vitro, immortalizing cells that are slow to proliferate, difficult to pass, and prone to senescence can provide additional cellular resources for in vitro disease studies [28]. Viral and cellular immortalization genes have been widely introduced to immortalize primary cells [29]. Among oncogenes, the *SV40LT* gene fragment is one of the most used targets for inducing cell immortalization. The integration of *SV40LT* into the nucleus of target cells leads to the binding and inactivation of P53 and Rb proteins, altering cell proliferation and prolonging the cell lifespan [30,31]. Stable cell line models are essential tools for studying the fundamental biological properties of tumors [32]. Owing to their convenience and cost-effectiveness, cell lines such as GIST882, GIST-T1, GIST430, GIST48, GIST62, and GIST552 are widely used in GIST studies [17,18,19]. However, immortalization methods for these cell lines have not been reported.

To the best of our knowledge, this is the first report of the successful establishment of a high-purity immortalized human GIST cell line (ImGIST), by transfecting primary GIST cells derived from a Chinese patient with *SV40LT*. In this study, ImGISTs were characterized using biomarkers, cytogenetics, and cytotoxicity experiments to describe their morphology, biological characteristics, genetic features, and comparative analyses with primary GIST cells. The culture condition for ImGISTs was optimized during cell culture to maximize the population doubling. The ImGIST cells had a significantly higher proliferation index and proliferation rate than the primary GIST cells. The ImGIST cells had a stable growth cycle, were passaged for more than 35 generations, and maintained a vigorous proliferation capacity. This cell line had the same genetic characteristics and immunophenotypes as the primary GIST cells. Approximately 85% of the GISTs have *C-KIT* or *PDGFRα* oncogenic mutations that constitutively activate downstream PI3K/AKT/mTOR and RAS/RAF/MEK pathways, which lead to cell proliferation and survival. *C-KIT* mutations are present in approximately 70–80% of the GISTs, with the most common mutation sites being in exons 11 (65–80%) and 9 (6–10%), followed by exons 13 (1–2%), 17 (<1%), and 8 (<1%) [10,33]. The benefit of recurrence-free survival after imatinib treatment depends on the location of the mutation in patients with GISTs, with those involving C-KIT 11 appearing to have the greatest benefit [34,35]. We detected an identical C-KIT 11 V560del mutation in ImGIST and primary cells, and follow-up drug sensitivity assays confirmed that the ImGIST cells were sensitive to imatinib. GISTs can acquire chromosomal aberrations, including deletions in chromosomes 14q, 22q, 1p, and 15q [36]. Typical deletions in chromosomes 14 and 22 were detected in ImGIST cells.

To establish cell culture models, it is important that these cells maintain the physiological and functional phenotype of the primary cells [37]. ImGISTs maximally maintain the biological and genomic characteristics of PriGISTs. The ImGIST cells had irregular neuronal cell-like radial growth and retained the characteristics of GIST cell fusion growth. Real-time PCR and western blotting confirmed the successful introduction of the *SV40LT* gene fragment into the primary cells and its stable expression in ImGIST cells. The reduction in P53 protein expression in ImGIST cells was consistent with the binding and inactivating of the *P53* gene by SV40LT to generate immortalized cell lines. As expected, ImGIST cells consistently and stably expressed CD117, which is a signature protein of GIST cells. The method of establishing immortalized GIST cell lines and the results obtained in this study are reproducible. The *SV40LT*-mediated immortalization has a specific reference value for the construction of other GIST cell models, which helps reduce researchers’ time and energy consumption and save experimental costs.

Current GIST cell line models cannot always fully encapsulate the morphology, gene expression patterns, and heterogeneity of human GISTs, nor can they preserve the immune microenvironment [15]. Currently, animal and organoid models are used as in vitro models for GIST research [38,39,40,41]. Organoids are structures constructed in 3D cultures of tumor tissues collected from patients. These organoids maintain the morphological structure of tumors and also their gene expression and heterogeneity [42]. Nevertheless, the use of organoids for studying GIST treatment is still in its initial stages [41], and future work will need to assess the biology of ImGISTs, induce drug resistance, investigate the mechanisms of secondary resistance, and explore their organoid formation capabilities.

To conclude, we successfully established an immortalized GIST cell line by introducing *SV40LT* into primary GIST cells. ImGISTs could stably express signature GIST proteins, maintain biological and genomic characteristics of normal primary GIST cells, and respond well to imatinib. The ImGIST cells could be a tool for studying the mechanisms of GIST tumorigenesis and progression and developing novel therapeutics.

## 4. Materials and Methods

### 4.1. Patient Clinical Information

This study was approved by the review committee of the Shengjing Hospital of China Medical University (approval number: 2022PS362K). GIST tissues were collected from the Shengjing Hospital of China Medical University. The patient was a 64-year-old untreated female with a preoperative diagnosis of GIST based on ultrasound endoscopy and computed tomography (Figure 7A,B). Open total resection was performed, and a postoperative diagnosis was made using HE staining, immunohistochemistry, and first-generation DNA sequencing. The pathological diagnosis was a gastric stromal tumor (modified National Institutes of Health risk classification: high risk). All relevant data and images were obtained with written informed consent from the patient after the purpose and nature of all procedures were explained in detail. All of this study’s procedures conformed to the provisions of the Declaration of Helsinki.

### 4.2. Isolation of Primary GIST Cells 

Immediately after surgical resection of the tumor, several tissue blocks (approximately 0.5 cm) were removed from the center of the tumor mass to avoid blood vessels. The tissue blocks were placed in RPMI1640 medium (iCell, Shanghai, China), sealed and stored at 4 °C, and transported to the laboratory within 1 h. Tumor tissues were washed three times with D-Hanks solution (Solarbio, Beijing, China) and dissected into 1 mm^3^ samples in a culture dish. The tissue was mixed with 5 mL type II collagenase (final concentration of 1 mg/mL; Solarbio), followed by digestion for 1.5 h in a 37 °C water bath (Figure 7C). Digestion was terminated by adding 5 mL culture medium and centrifugation at 1000 rpm for 5 min. The cell suspension was filtered twice through a 200-mesh filter. Finally, the RPMI1640 complete medium was added, and the cell suspension was transferred to T25 culture flasks. The complete medium was supplemented with 15% FBS (iCell), 1% penicillin-streptomycin solution (iCell), 1× L-glutamine (iCell), and 20 ng/mL basic fibroblast growth factor (bFGF, iCell). During the culture period, cells were incubated in a humidified incubator at 37 °C with 5% CO_2_, and the medium was changed every 2 d. The primary cell line was named PriGIST, and the PriGIST cells were divided into two parts: one was introduced with genes for immortalization, and the other was used as a control.

### 4.3. Immortalization of Primary GIST Cells

The SV40LT was purchased from Cell Signaling Technology (Danvers, MA, USA). PriGIST cells (P0) were divided into two groups that were cultured to a logarithmic growth phase in 24-well plates. The first group of cells was used to determine puromycin concentration and SV40LT (20 μL/mL) was added into the culture medium of the second group of cells as directed. The cells were cultured in the virus-containing medium for 12 h before being replaced with fresh medium. After 24 h incubation, cells with stable viral integration were selected with 1 μg/mL puromycin, and the selection medium was changed every 48 h after that. The selected cells were used for amplification. Immortalized GIST cells were named ImGISTs.

### 4.4. Cell Morphology 

Morphological observations of ImGIST and PriGIST cells were performed using a light microscope (Nikon, Minato-ku, Tokyo, Japan).

### 4.5. Immunofluorescence Identification

PriGIST cells in the logarithmic growth phase were seeded at a density of 2 × 10^4^ cells/well in a 12-well plate with preplaced crawl sheets. Immunofluorescence staining was performed at 80% cell density after approximately 12 h of culture. Cells were fixed in 4% paraformaldehyde (Elabscience, Wuhan, China) for 30 min and washed three times in PBS (Procell, Wuhan, China). The cells were permeabilized with 0.1% Triton X-100 (Elabscience) for 20 min and then incubated in 5% bovine serum albumin (BSA; Elabscience) for 2 h to block the antigen. Primary antibodies for DOG-1 (1:100 dilution, ZSGB-BIO, Beijing, China), CD34 (1:100 dilution, ZSGB-BIO), and CD117 (1:100 dilution, ZSGB-BIO) were added and incubated overnight at 4 °C. Goat anti-rabbit IgG (H + L) (1:100 dilution; Elabscience) was used as the secondary antibody. The cell nuclei were re-stained with 4′,6-diamidino-2-phenylindole (DAPI, Solarbio). Images were captured using a confocal microscope (Olympus Corporation, Tokyo, Japan).

### 4.6. Genetic Mutation Analysis

The ImGIST and PriGIST cells were tested for C-KIT and PDGFRα mutations. The templates were amplified according to the manufacturer’s instructions (Sinomdgene, Beijing, China) and the results were analyzed using an Applied Biosystems 7500 Real-Time PCR System.

### 4.7. Short Tandem Repeat Analysis

The PriGIST cells were used as control samples and ImGIST cells were identified using the short tandem repeat (STR) analysis. The DNA was extracted using a commercial kit from CORNING (AP-EMN-BL-GDNA-250G). Twenty-one STRs, including the amelogenin locus, were amplified using the PowerPlex 21D System (iCell) and separated using an ABI 3730XL Genetic Analyzer (Thermo Fisher Scientific Inc., Waltham, MA, USA). Signals were analyzed using GeneMapper ID V. 32.

### 4.8. Karyotype Analysis 

The ImGIST cells were treated with 0.25 μg/mL colchicine (Sigma-Aldrich, St. Louis, MO, USA) for 1–2 h at 37 °C, followed by cell collection. Cells were incubated in a hypotonic solution for 30 min at 37 °C and then fixed in methanol/glacial acetic acid (*v*/*v* = 3:1) for 10 min. The cells were digested with trypsin, fixed, and then stained with Giemsa stain (Solarbio). Mitotic cells with good dispersion and moderate staining were selected for karyotypic observation and microscopic analysis.

### 4.9. Cell Growth Assay

The CCK-8 assay was used to compare the growth curves of ImGIST and PriGIST cells. The two cell types in the logarithmic growth phase were seeded in 96-well plates at a density of 8 × 10^3^ cells/well in 100 μL complete medium. The cells were cultured for 1–7 d, and the medium was renewed every 2 d. Six replicate wells were sampled every 24 h by removing the medium and replacing it with 90 μL of serum-free medium and 10 μL of CCK-8 solution (Beyotime, Shanghai, China). Blank controls were also prepared, and the plates were incubated in a cell incubator for 2 h. Absorbance (OD value) was measured at 450 nm using a microplate reader (BioTek, Winooski, VT, USA). The Td was calculated using the following formula: Td = t × lg2/lg (Nt/N0). The experiment was repeated three times, and the data were analyzed using GraphPad Prism software 7.00. PriGISTs were used as the controls.

### 4.10. Cell Cycle 

ImGIST and PriGIST cells were washed with pre-cooled PBS and fixed overnight in 70% ethanol at 4 °C. The next day, cells were resuspended in 500 μL propidium iodide/RNase A staining solution (Beyotime) and then warmed to 37 °C for 30 min in the dark. The cell cycle was detected using flow cytometry (Miltenyi Biotec, Bergisch Gladbach, Germany). The proliferation index (PI) was calculated as follows: PI = (S + G2/M)/(G0/G1 + S + G2/M).

### 4.11. Real-Time PCR

Total mRNA was extracted from the two cell types using RNAeasyTM Plus (Beyotime), and cDNA was synthesized using ReverTra Ace-α- (TOYOBO). Real-time PCR was performed according to the manufacturer’s instructions (Takara, Dalian, Japan). The cycle conditions for the PCR were as follows: 95 °C, 10 s; 40 cycles of 95 °C, 5 s, 60 °C, 30 s; 95 °C, 15 s, 60 °C, 60 s, 95 °C, 15 s. Data were analyzed with the 2^−ΔΔCt^ value calculation using glyceraldehyde 3-phosphate dehydrogenase (*GAPDH*) for normalization. Primer sequences are listed in Table 2.

### 4.12. Western Blot Analysis

The PiGIST and ImGIST cells were collected and processed on ice. The cells were lysed using a radioimmunoprecipitation assay (RIPA, Solarbio) containing phosphatase and protease inhibitors (Solarbio), and the supernatant was collected after centrifugation. Next, 5× loading buffer (Solarbio) was added to the supernatants, and the mixtures were boiled at 100 °C for 10 min in a water bath. Protein samples (40 μg) were separated by 12% sodium dodecyl sulfate-polyacrylamide gel electrophoresis and transferred to polyvinylidene fluoride membranes (Millipore, Billerica, MA, USA). After blocking with 5% BSA, the membranes were incubated overnight at 4 °C with primary antibodies against CD117 (1:1000 dilution; Proteintech, Wuhan, China), SV40 (1:1000 dilution; Proteintech), P53 (1:1000 dilution; Proteintech), and GAPDH (1:10,000 dilution; Proteintech). The membranes were then incubated with goat anti-rabbit secondary antibodies (1:10,000 dilution, Proteintech) or goat anti-mouse secondary antibodies (1:10,000 dilution, Proteintech) at 37 °C for 30 min. Protein expression was visualized using an enhanced chemiluminescence reagent (Tanon, Shanghai, China). The results were semi-quantified using ImageJ software 1.46r (National Institutes of Health, Bethesda, MD, USA) [43] and normalized to that of GAPDH.

### 4.13. Drug Sensitivity Test

The CCK-8 assay was used to determine the inhibitory effect of imatinib (Sigma-Aldrich) on the proliferation of the immortalized GIST cells. ImGIST cells in the logarithmic growth phase were seeded at a density of 8 × 10^3^ cells/well in 96-well plates in 100 μL complete medium. The medium was replaced with imatinib-containing medium after 24 h using a drug concentration gradient of 0, 5, 10, 25, 50, and 100 μM imatinib. The media were removed at 24 and 48 h, and 90 μL of serum-free medium and 10 μL of CCK-8 solution were added to each well. Blank controls were also prepared and incubated in a cell incubator for 2 h. Absorbance (OD) was measured at 450 nm using a microplate reader. The experiment was repeated three times, and data were analyzed using GraphPad Prism software and used to calculate the half-maximal inhibitory concentration (IC50).

### 4.14. Effects of Serum Concentration on Cell Growth

The CCK-8 assay was used to determine the optimal serum concentration for ImGST cell cultures. ImGIST cells were resuspended at a density of 8 × 10^4^ cells/mL in medium containing 0%, 5%, 10%, 5%, and 20% FBS, respectively. Cells were seeded in 96-well plates. The media were removed after 24 and 48 h and replaced with 90 μL of serum-free medium and of 10 μL CCK-8 solution. Blank controls were also prepared and incubated in a cell incubator for 2 h. Absorbance (OD) was measured at 450 nm using a microplate reader. The experiment was repeated three times.

### 4.15. Statistical Analysis

All statistical analyses were performed using the GraphPad Prism 7.00 software (GraphPad Software, San Diego, CA, USA). The data were acquired from three independent experiments and presented as mean ± standard deviation (SD). All data were evaluated using a one-way analysis of variance followed by Tukey’s multiple comparison test, with the threshold for statistical significance set at *p* < 0.05.

## Figures and Tables

**Figure 1 ijms-24-13640-f001:**
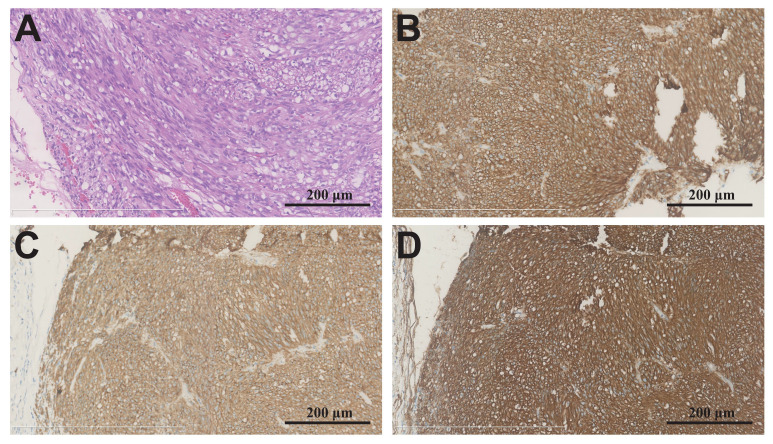
Pathological assessment of GIST tissue. (**A**) Hematoxylin-eosin (HE) staining of tumor tissue. The GIST tissue positively stained for (**B**) DOG-1, (**C**) CD117, and (**D**) CD34. Magnification = 200×.

**Figure 2 ijms-24-13640-f002:**
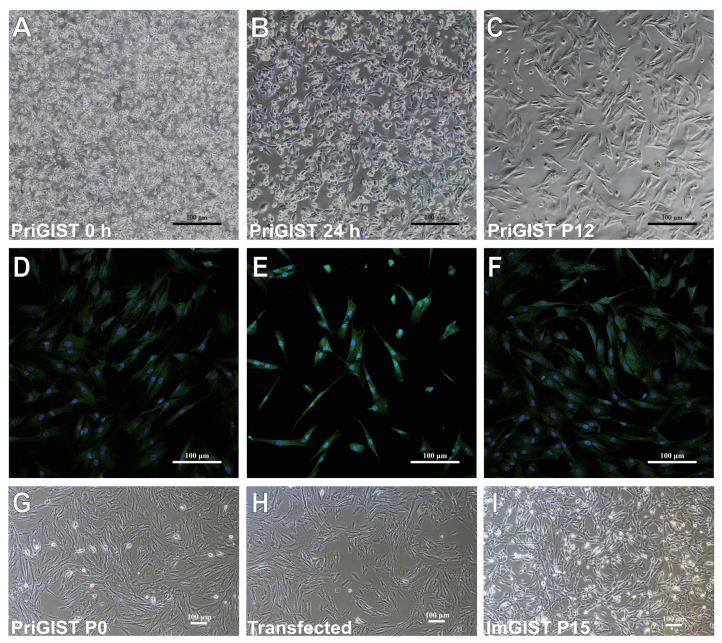
Protein expression and cell morphology of PriGIST and ImGIST cells. (**A**) Isolation of PriGIST cells (100×). (**B**) Morphological observations of PriGIST cells after 24 h of adherence (100×). (**C**) Morphological observations of PriGIST cells at P12 (100×). Immunofluorescence staining showed PriGIST cells positively expressed (**D**) DOG-1, (**E**) CD117, and (**F**) CD34 (100×). Morphological observation of (**G**) PriGIST cells (P0), (**H**) PriGIST cells during transfection, and (**I**) ImGIST cells (P15) (100×). PriGIST cells: primary GIST cells; ImGIST cells: immortalized GIST cells; P: passage.

**Figure 3 ijms-24-13640-f003:**
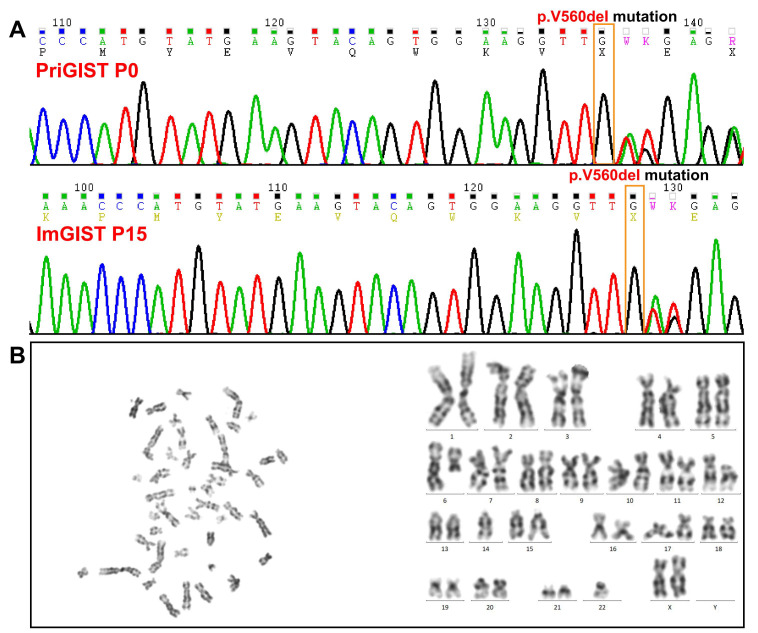
Cytogenetic characteristics of ImGIST cells. (**A**) Sanger sequencing detected identical *C-KIT* 11 V560del mutations in ImGIST and PriGIST cells (yellow box). (**B**) The representative karyotype of ImGIST cells was 44–45, XX. +3, del(6)(q13), der(6;12)(p10;q10), add(7)(p13), del(12)(p11), add(12)(p11), −14, −22.

**Figure 4 ijms-24-13640-f004:**
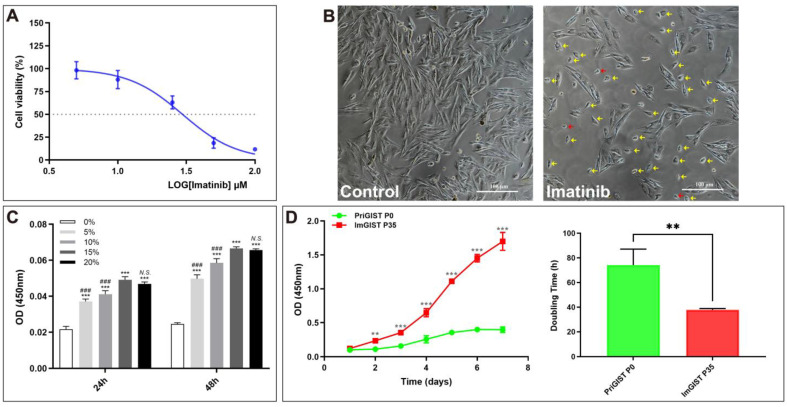
Biological characteristics of ImGIST cells. (**A**) Imatinib treatment inhibited ImGIST cell growth (**B**) After treatment with imatinib, most ImGIST cells had a small spindle-shaped morphology and lost their fusion growth ability (yellow arrow), while a few cells died (red arrow, 100×). (**C**) The optimal fetal bovine serum concentration for the maintenance culture of ImGIST cells was 15%. *** *p* < 0.001, vs. 0% group; ### *p* < 0.001, vs. 15% group; *N.S.*, no significant difference, vs. 15% group. (**D**) The cell growth curves were determined using the CCK-8 assay between ImGIST and PriGIST cells. The mean Td values for ImGIST and PriGIST cells were 37.8 and 74.1 h, respectively. ** *p* < 0.01, vs. PriGIST P0 group; *** *p* < 0.001, vs. PriGIST P0 group. IC50: half maximal inhibitory concentration; Td: cell population doubling time.

**Figure 5 ijms-24-13640-f005:**
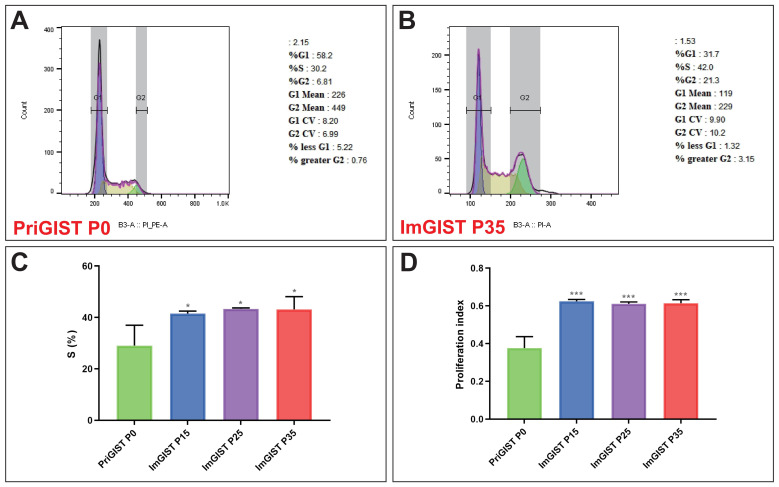
Cell cycle of PriGIST cells and ImGIST cells. Cycle distribution of (**A**) PriGIST and (**B**) ImGIST cells. (**C**) The percentage of PriGIST and ImGIST cells in S phase of the cell cycle. (**D**) The proliferation indices of ImGIST (P35) and PriGIST cells. All data are presented as means ± SD. *** *p* < 0.001, vs. PriGIST P0 group; * *p* < 0.05, vs. PriGIST P0 group.

**Figure 6 ijms-24-13640-f006:**
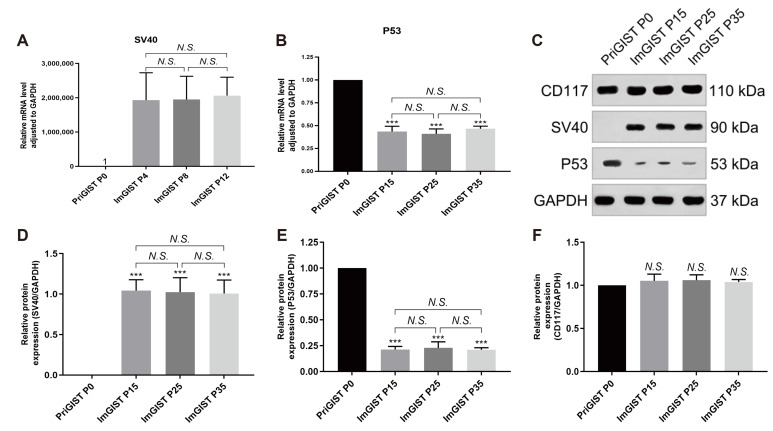
Expression of SV40, P53, CD117, and GAPDH in PriGIST and ImGIST cells was measured by real-time PCR and western blotting. (**A**,**B**) Real-time PCR was used to detect the mRNA levels of *SV40* and *P53* in PriGIST and ImGIST cells normalized to *GAPDH* (*n* = 4). (**C**) Representative image of a western blot showing bands for SV40, P53, CD117, and GAPDH. (**D**–**F**) Relative optical density was normalized to that of GAPDH (*n* = 4). All data are presented as means ± SD. *** *p* < 0.001, vs. PriGIST P0 group; *N.S.*, no significant difference when compared to PriGIST P0 in panel F. SV40: Simian virus 40; P: passage.

**Figure 7 ijms-24-13640-f007:**
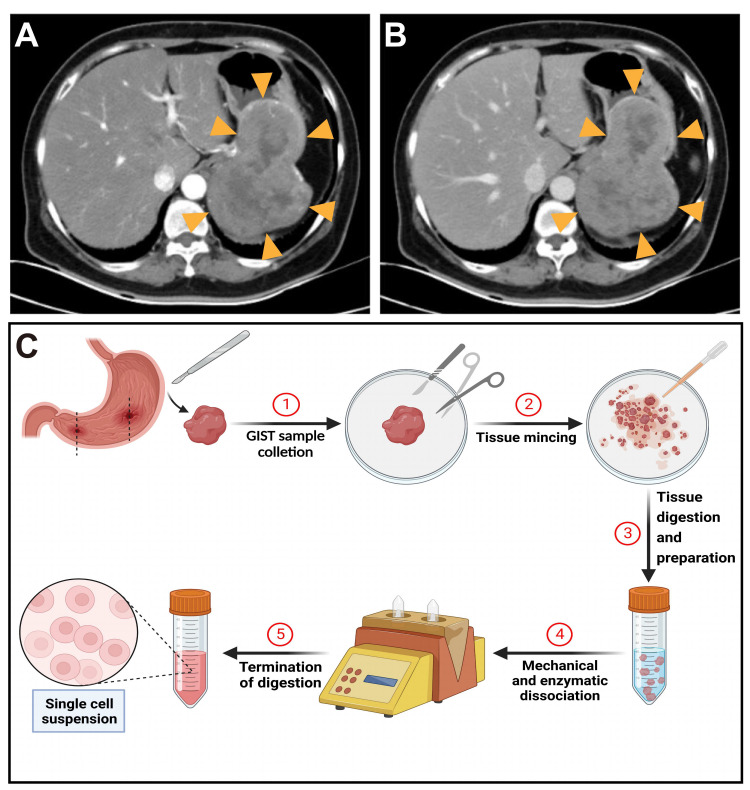
Clinical imaging information of the patient and the isolation of primary GIST cells. Abdominal computed tomography images of the patient in the (**A**) arterial phase and (**B**) venous phase. The lesion site is indicated by yellow triangles. (**C**) Digestion and isolation process of primary GIST cells.

**Table 1 ijms-24-13640-t001:** Short tandem repeat analysis results showing the match between ImGIST and PriGIST cells.

Locus	PriGIST	ImGIST
Amelogenin	X, X	X, X
D5S818	10, 12	10, 12
D13S317	8, 8	8, 8
D7S820	8, 13	8, 13
D16S539	11, 12	11, 12
VWA	14, 14	14, 14
TH01	9, 10	9, 10
TPOX	8, 8	8, 8
CSF1PO	10, 11	10, 11
D12S391	19, 20	19, 20
FGA	23, 24	23, 24
D2S1338	22, 24	22, 24
D21S11	29, 30	29, 30
D18S51	17, 17	17, 17
D8S1179	10, 11	10, 11
D3S1358	15, 17	15, 17
D6S1043	12, 12	12, 12
PENTAE	12, 20	12, 20
D19S433	16, 17.2	16, 17.2
PENTAD	10, 11	10, 11
D1S1656	11, 16	11, 16

**Table 2 ijms-24-13640-t002:** Primer sequences for real-time PCR.

Genes	Sequences
*SV40*	Forward: 5′-TTC AGA GCA GAA TTG TGG AGT G-3′
	Reverse: 5′-CCT GGCT GTC TTC ATC ATC ATC-3′
*P53*	Forward: 5′-CCA ACA ACA CCA GCT CCT CT-3′
	Reverse: 5′-CCT CAT TCA GCT CTC GGA AC-3′
*GAPDH*	Forward: 5′-AGT CCA CTG GCG TCT TCA-3′
	Reverse: 5′-AGG CTG TTG TCA TAC TTC TCA T-3′

SV40: Simian virus 40; GAPDH: glyceraldehyde 3-phosphate dehydrogenase.

## Data Availability

The data presented in this study are available in the article.

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
