# Peer review of "Characterization of a Human Gastrointestinal Stromal Tumor Cell Line Established by SV40LT-Mediated Immortalization"

_ijms, 2023, doi:10.3390/ijms241713640_

Round 1

Reviewer 1 Report

Bargraphs must be presented as bar with dot plots.

Discuss more about the significance of this work.

There is a lot of scope to improve the language. Must be checked by some English language expert.

Author Response

Response to Reviewer 1 Comments

Thank you very much for your comments on our manuscript. We have revised the manuscript and moderated the bar graphs according to your detailed suggestions.

Point 1: Bar graphs must be presented as bar with dot plots.

Response 1: We have moderated the bar graphs by adding dot plots. And groups without dot plots indicate controls used for normalization.

Point 2: Discuss more about the significance of this work.

Response 2: We have added some discussion about the significance of the work in the discussion section. (Page 8, Line 211,212; Page 8, Line 229,230; Page 8, Line 237-240; Page 9, Line 254,255).

Reviewer 2 Report

1. Overall, the Authors explained the study well, but you can't strongly support the theory of this paper. In this study, the authors used only one patient-derived cell line. The genetic characteristics of the two GIST patients are different. So we can't completely agree. It is better if they use multiple cell lines for at least a couple of experiments, and if they don't see the difference, they can continue with one cell line for the rest. 

2. Is there any disadvantages of using SV40LT transduction?

Author Response

Response to Reviewer 2 Comments

Thank you very much for your recognition of our review and thank you for pointing out the important limitations.

Point 1: Overall, the Authors explained the study well, but you can't strongly support the theory of this paper. In this study, the authors used only one patient-derived cell line. The genetic characteristics of the two GIST patients are different. So we can't completely agree. It is better if they use multiple cell lines for at least a couple of experiments, and if they don't see the difference, they can continue with one cell line for the rest.

Response 1: Thank you very much for pointing out the important limitations. Early in our research, we set out to create a GIST cell line capable of spontaneous immortalization. However, several experiments have shown that primary GIST cells exhibit drastic morphological changes and growth retardation during in vitro culture for approximately 10 generations and cannot undergo spontaneous immortalization. Due to the failure of spontaneous immortalization and the scarcity of clinical tissue sources at that time, we selected only a group of recently isolated and cultured primary GIST cells with clear backgrounds for SV40LT-mediated immortalization and subsequent experiments. We all agree that a couple of experiments are needed to validate the stability of SV40LT-mediated immortalization, but we do not have enough clinical tissue sources for the experiments for now. In the future, we will perform more isolation and culture of primary GIST cells to verify the stability of SV40LT-mediated immortalization.

Point 2: Is there any disadvantages of using SV40LT transduction?

Response 2: To the best of our knowledge, SV40LT transduction has certain disadvantages that do not affect the subsequent use of the cell models: (1) The transfected cells may undergo particular changes in their biological properties, such as changes in cell morphology. In our study, the ImGISTs cells had a shorter spindle-shaped morphology than that of the PriGISTs cells and irregular neuronal cell-like radial growth, while the fused growth characteristic associated with PriGISTs remained. (2) The transfected cells may undergo certain alterations in their genetic profile. Keratinocyte cell lines immortalized by SV40LT had several undesirable genetic abnormalities, such as mutations in P53 or aneuploidy isochromosomes (Lehman et al., 1993; Allen-Hoffmann et al., 2000). However, SV40LT transduction also has disadvantages that affect the subsequent use of the cell models. For example, SV40LT inactivates the p300/CBP (cAMP-response element-binding protein) family of transcriptional coactivators, resulting in the aberrant differentiation of pre-adipocytes after immortalization and the loss of adipogenic capacity (Darimont et al., 2003).

Reviewer 3 Report

Gastrointestinal stromal tumor (GIST) cell lines are widely used as in vitro models for experimental studies. As Xiangchen Hu et al. showed, the immortalization methods of these cell lines are unknown. Thus, they used simian virus 40 large T antigen (SV40LT) to transfect primary GIST (PriGIST) cells to establish an immortalized human GIST cell line (ImGIST).

The whole procedure is adequately documented. ImGIST cell line was characterized by comparing cell morphology, analysis of genetic mutations and short tandem repeats pattern, and karyotype analysis. Moreover, the biological characteristics of ImGIST and PriGIST cell lines were compared. ImGIST cells had a significantly higher proliferation index and proliferation rate than primary GIST cells.

In my opinion, the manuscript is well-written, and the quality of the experiments is high.

I only found mistakes in references citing style - the journal name should be present in abbreviation form and not full name, e.g., 

The New England journal of medicine should have the format N. Engl. J. Med.

Author Response

Response to Reviewer 3 Comments

Thank you very much for your comments on the references citing style.

Point 1: The New England journal of medicine should have the format N. Engl. J. Med.

Response 1: We have changed all the journal names from full names to abbreviation forms in our manuscript according to your detailed suggestions.

Round 2

Reviewer 2 Report

I have no further comments.